# Progressive alterations in mineral contents in citrus genotypes toward *Alternaria citri* causing brown spot of citrus

Shahid Iqbal[1,2], Muhammad Atiq[1]*, Shahbaz Talib Sahi[1], Nadeem Akbar[3], Nasir Ahmed Rajput[1]

1 Department of Plant Pathology, University of Agriculture Faisalabad, Faisalabad, Pakistan, 2 National Agriculture Research Center, Crop Disease Research Institute, PARC, Islamabad, Pakistan, 3 Department of Agronomy, University of Agriculture Faisalabad, Faisalabad, Pakistan

* muhammad.atiq@uaf.edu.pk

**Data Availability Statement:** I have uploaded data of manuscript on the website of the Journal as Supporting information.

## Abstract

Brown spot of citrus caused by *Alternaria citri* is one of the emerging threats to the successful production of citrus crops. The present study, conducted with a substantial sample size of 50 leaf samples for statistical reliability, aimed to determine the change in mineral content in citrus leaves after brown spot disease attack. Leaf samples from a diverse range of susceptible citrus varieties (Valentia late, Washington navel, and Kinnow) and resistant varieties (Citron, Eruka lemon, and Mayer lemon) were analyzed. Significant variations (p ≤ 0.05) in mineral contents were observed across reaction groups (inoculated and un-inoculated), types (resistant and susceptible), and varieties of citrus in response to infection of *Alternaria citri*. The analysis of variance showed significant changes in mineral levels of citrus leaves, including nitrogen (N), phosphorus (P), potassium (K), calcium (Ca), magnesium (Mg), zinc (Zn), sodium (Na), iron (Fe), and copper (Cu). The results indicate that the concentration of N and P differed by 6.63% and 1.44%, respectively, in resistant plants, while susceptible plants showed a difference of 6.07% and 1.19%. Moreover, resistant plants showed a higher concentrations of K, Ca, Mg, Zn, Na, Fe, and Cu at 8.40, 2.1, 1.83, 2.21, 1.58, 2.89, and 0.36 ppm respectively, compared to susceptible plants which showed concentrations of 5.99, 1.93, 1.47, 1.09, 1.24, 1.81, and 0.31 ppm respectively. Amounts of mineral contents were reduced in both resistant as well as susceptible plants of citrus after inoculation. Amount of N (8.56), P (1.87) % while K (10.74), Ca (2.71), Mg (2.62), Zn (2.20), Na (2.08), Fe (3.57) and Cu (0.20) ppm were recorded in un-inoculated group of citrus plants that reduced to 3.15 and 0.76% and 3.66, 1.40, 0.63,0.42, 0.74, 1.13 and 0.13 ppm in inoculated group respectively. It was accomplished that susceptible varieties contained lower ionic contents than resistant varieties. The higher concentrations of ionic contents in resistant citrus varieties build up the biochemical and physiological processes of the citrus plant, which help to restrict spread of pathogens. Further research could explore the interplay between mineral nutrition and disease resistance in citrus, potentially leading to the development of new disease-resistant varieties.

**Funding:** The author(s) received no specific funding for this work.

**Competing interests:** The authors have declared that no competing interests exist.

## Introduction

Citrus is considered as an imperative fruit crop across the world that belongs to family Rutaceae and subfamily Aurantioideae [1]. It is originated from sub-tropical and temperate areas of the South East Asia [2]. It contributes 1.6% in Pakistan's GDP. It ranked 1st in terms of production among all fruits and Pakistan exports 0.8% of the total production. It is regarded as most valuable fruit as it contains large amount of amino acids, sugars, organic acids, vitamins, antioxidants like terpenes, flavonoids, macro as well as micronutrients minerals (magnesium and calcium) along with rich source of ascorbic acid [3, 4]. However, the production of citrus is affected by various biotic and abiotic factors. In biotic factors different fungal, bacterial, nematodes and viral diseases are found to be destructive to citrus crop [5–7]. Among fungal diseases, brown spot which is caused by *Alternaria citri* is an emerging threat to the successful production of citrus [8]. It causes a huge economic loss in different citrus growing areas of the world [9]. Circular lesions (brown to black coloured), twig blight, dropping of leaves and softening of fruits are the characteristic symptoms of brown spot of citrus but when conditions become favorable, wilting of shoot occur that leads to the death of the whole plant. On fruits, lesions differ from minute spot to large crater like lesions. Lesions on leaves and twig blight are the primary source for conidial germination and maximum conidia are produced in the presence of moisture.

Mineral nutrients are not only essential for plant growth but also have a key role in plant-pathogen interaction. These nutrients minimize disease to a tolerable level but any interruption in ionic contents retards the defense mechanism of plant resulting in disease development. These nutrients are not responsible for disease resistance but they are also involved in the development of disease thereby fluctuating the functions of plants [10]. Similarly, these nutrients influence disease susceptibility through induction of chemical alterations in the host plants. Moreover, nutrients play a key role in the defense system of plants. These can be categorized into four groups on the basis of biochemical and physiological behavior [11]. The main elements like C, H, O and N are involved in enzymatic and oxidation-reduction reactions. The second group (B and P) take part in energy transfer reactions while elements of third group (K, Ca, Mg, Mn and Cl) act as catalyst and fourth group minerals are known as structural chelates [12, 13]. But when citrus plant is attacked by *Alternaria citri*, it causes various changes in its metabolism, biochemical compounds and disrupt the nutrient uptake, absorption, mobility, assimilation, and nutrient consumption or translocation is influenced by pathogens thereby resulting in nutrient deficiencies. These nutrients have an important effect on all key components which are involved in disease severity. Moreover, plants have maximum chances of damage by pathogens that contained less amount of nutrients that are essential for buildings cell wall as well as other structural tissues [14]. Similarly, [15] reported that pathogens, after attack, competes with host plants for Na, Mg, Ca, K, Cl and $H_2PO_4$ to initiate infection by increasing or decreasing their amounts in the plant. This alteration affects the plant's health and vigor. Most of fungi are very destructive as they reduce the mineral uptake efficiency of plants and imbalance of these mineral nutrients cause physiological as well as metabolic problems thereby enhancing susceptibility in plants towards the diseases [16]. This study aimed to identify alterations in mineral nutrients in citrus plants following pathogen attacks, contributing to the successful management of citrus brown spot. To achieve this objective, mineral profiling of inoculated and un-inoculated citrus plants of susceptible as well as resistant cultivars was conducted.

## Materials and methods

One-year-old plants of six varieties including resistant (Valentia late, Washington navel, Kinnow) and susceptible types (Mayer lemon, Eruka lemon and Citron) were collected from the

fruit nursery, Institute of Horticultural Sciences (IHS), University of Agriculture Faisalabad (UAF). All varieties were planted in three replications in pots (32×22cm) that contained field soil previously sterilized by drenching a 1% solution of formalin under Completely Randomized Design (CRD) in the greenhouse at Research Area Department of Plant Pathology, University of Agriculture Faisalabad (UAF). The greenhouse conditions were maintained at a relative humidity (RH) of approximately 70% and a temperature ranging between 25 to 30°C. These conditions were closely monitored using digital hygrometers and thermometers throughout the experimental duration to ensure consistency and optimal growth conditions for citrus plants. All the horticultural practices were followed including doses of fertilizers (Each pot received a balanced fertilizer consisting of 10 grams of 20-20-20 NPK per square meter of soil surface area) and number of irrigations (once per week, in line with the controlled environment of the greenhouse). to keep the plants in a healthy condition. Diseased samples showing characteristic symptoms of brown spot of citrus were collected in brown paper bags (13" × 9.5") after one month of planting and brought to the plant pathological Laboratory in order to isolate pathogen associated with brown spot of citrus. For the isolation of pathogen, PDA media was used. For the pathogenicity test, a conidial suspension was prepared by adding sterile distilled water to Petri plates and scraping the agar surface with a sterilized blade. After filtering through sterile muslin cloth, suspension was adjusted to $1 \times 10^4$ conidia/ mL using a heamocytometer (Model: T20B05) [17]. For inoculation, the fungal suspension was taken in a sterile syringe (23Gx 1") and injected into the midrib and veins of lower surface of leaves [18, 19] early in the morning (when the maximum number of stomata were opened). Disease incidence was confirmed by Koch's postulates after one month of incubation period. Samples were collected from susceptible and resistant citrus plants from both inoculated and un-inoculated groups for the estimation of mineral contents.

## Sample preparation for the analysis of mineral contents

Leaves samples were collected and brought to citrus pathology Laboratory for further process. For the analysis of mineral contents, samples were dried in an oven (Memmert: UN 30) at 65°C for 4 hrs as described in previous investigations. To obtained a fine powder, samples were grinded with pestle and mortar and 0.5g of each sample was subjected to boiling in 10mL 1.4N nitric acid ($HNO_3$) at 100°C for 30 minutes on a hotplate (RT2-230V). After cooling, the suspension was diluted with distilled water and examined for the estimation of mineral contents [20, 21]. The data was analyzed using Nested structure design.

## Estimation of phosphorus (P) from the inoculated and un-inoculated citrus plant leaves

A sample solution of 0.1mL was prepared through wet dilution method and poured into a volumetric flask (ASTM-E288). Then, 1mL of ammonium molybdate [$(NH_4)_6Mo_7O_{24}$] and 8.6mL of distilled water was taken and poured into a volumetric flask (250mL). Solution was mixed by stirring and 0.4mLof naphthol sulphonic acid ($C_{10}H_8O_4S$) was added. Distilled water was used as a substitute for the sample solution serve as a control. Absorbance was taken at a wavelength of 720nm through Spectrophotometer (Model 121–003). Absorbance was compared to standard curve [22] using an atomic absorption spectrometer (Hitachi, polarized Zeeman) to determine the concentration of phosphorus in both inoculated and uninoculated citrus leaves.

### Estimation of potassium (K) and sodium (Na) from inoculated and uninoculated leaves of citrus plants

Determination of ionic contents (Sodium and Potassium) was done using a Flame photometer (Model: PFP-7) following the official methods of analysis [23]. For the determination of Sodium (Na) and potassium (K), Sodium Chloride (NaCl) and Potassium Chloride (KCl) was taken as a standard and fresh working standard was prepared in order to obtain the Sodium (Na) and Potassium (K) standards curves by using four concentrations (10. 20, 30 and 40ppm) for both Na and K.

### Estimation of magnesium (Mg), calcium (Ca) copper (Cu), zinc (Zn) and iron (Fe) from the inoculated and un-inoculated leaves of citrus

Determination of magnesium, copper, iron, calcium, and zinc was done using Atomic Absorption Spectrometer (AAS) (Hitachi, polarized Zeeman) by using standards such as magnesium sulphate ($MgSo_4$), copper sulphate ($CuSO_4$), calcium chloride ($CaCl_2$) iron sulphate ($FeSO_4$) and zinc sulphate ($ZnSO_4$). Standard curves were determined through concentrations of Mg (5, 10, 15 and 20 ppm), Cu (2, 2.5, 3 and 3.5 ppm), Ca (10, 20, 40, 80 and 100 ppm), Fe (1, 2, and 3 ppm) and 0.2, 0.3, 0.5 and 2 ppm for zinc, obtained as their fresh working standards just before this activity.

### Estimation of nitrogen/ crude protein from the inoculated and un-inoculated leaves citrus plants

Crude nitrogen contents were estimated from the collected samples by using Kjeldahl method [24]. After this, 5g of digestion mixtures (containing potassium sulphate and copper sulphate) and 25 mL of concentrated $H_2SO_4$ were added to necked flask. The sample was subjected to boiling in a digestion hood (KB8S Kjeldatherm) to achieve the clear contents. Then these contents were cooled, placed in a 250mL volumetric flask (ASTM-E288) and were diluted with distilled water. 10mL solution was poured in a distillation apparatus (Model: VAP20) of micro Kjeldahl in 10 mL of 40% NaOH solution. In this reaction, ammonia (NH3) was formed and was placed in a beaker which contained 10mL of 2% $H_3BO_3$ and 2–3 drops of methyl red that act as an indicator. After a while, solution was exposed to sulfuric acid ($H_2SO_4$) as 0.1 N standard to make pink point and nitrogen (N) was calculated by using formula:

$$\text{Nitrogen (\%)} = \frac{0.1 \text{ N H2SO4} \times 0.0014 \times 250 \times 100}{\text{WI} \times 100}$$

The sample crude protein sample %age was calculated by the givena formula

$$\text{Crude Protein} = \text{Nitrogen (\%)} \times 6.25$$

### Statistical analysis

The plant samples were divided into two groups: inoculated and un-inoculated and each group further contained two reaction types: i) Susceptible ii) Resistant, three susceptible varieties: Valentia late, Washington navel, Kinnow and three resistant varieties: Mayer lemon, Eruka lemon and Citron were used. For the estimation of mineral contents, standard analytical methods via Nested Structure Design were used and data was statistically analyzed through PROC MIXED procedure of the statistical analysis system [25].

## Results

### Estimation of minerals contents of N, P, K, Na, Zn, Ca, Mg, Fe and Cu from inoculated and un-inoculated citrus plant

Significant alterations were recorded among un-inoculated (8.56%) and inoculated (3.15%) leaves of citrus plants, explaining that nitrogen content affects the metabolic activities of citrus plants after the attack of brown spot disease (Table 2).Varieties categorized as resistant and susceptible also indicated significant variation, with a total variance of 93.02% at P<0.05 (Table 1). Individually, Mayer lemon expressed the highest concentration of nitrogen at 6.95%, and Washington navel showed 5.85% after the analysis (Table 2). In the case of phosphorus, a significant difference was noticed among inoculated and un-inoculated citrus plant leaves (on an average 0.76 and 1.87%, respectively) under diseased conditions, with a total variance of 93.49% (Tables 1 and 2). Resistant (1.44%) and susceptible plants (1.19%) also indicated significant variation at P<0.05 (Table 1). 1.34% of total variance was indicated by cultivars in their natural affinities with respect to phosphorus contents. Mayer lemon and Valentia late expressed highest as well as lowest P concentration to extent of 1.45 and 1.09%

**Table 1. Nested structured ANOVA for N, P, K, Ca and Mg contents of inoculated and un-inoculated citrus plant leaves.**

| Nitrogen (%) | | | | | | | |
|---|---|---|---|---|---|---|---|
| SOV | DF | SS | MS | F value | Pr>F | Variance components | % of total variance component |
| Group (A) | 1 | 839.455 | 839.455 | 29.707 | 0.032* | 15.022 | 93.02 |
| Type (B) | 2 | 56.516 | 28.258 | 27.104 | 0.000* | 1.008 | 6.24 |
| Variety (C) | 8 | 8.340 | 1.042 | 327.461 | 0.000* | 0.115 | 0.72 |
| Error | 96 | 0.305 | 0.003 | - | - | 0.003 | 0.02 |
| Total | 107 | 904.618 | - | - | - | 16.149 | - |
| Phosphorus (%) | | | | | | | |
| Group (A) | 1 | 34.307 | 34.307 | 35.081 | 0.027* | 0.617 | 93.49 |
| Type (B) | 2 | 1.955 | 0.977 | 12.170 | 0.004* | 0.033 | 5.04 |
| Variety (C) | 8 | 0.642 | 0.080 | 89.080 | 0.000* | 0.009 | 1.34 |
| Error | 96 | 0.086 | 0.000 | - | - | 0.001 | 0.14 |
| Total | 107 | 36.992 | - | - | - | 0.660 | - |
| Potassium (ppm) | | | | | | | |
| Group (A) | 1 | 1349.945 | 1349.945 | 17.137 | 0.054* | 23.540 | 88.79 |
| Type (B) | 2 | 157.544 | 78.772 | 114.811 | 0.000* | 2.892 | 10.91 |
| Variety (C) | 8 | 5.488 | 0.686 | 246.034 | 0.000* | 0.076 | 0.29 |
| Error | 96 | 0.267 | 0.002 | - | - | 0.012 | 0.053 |
| Total | 107 | 1513.246 | - | - | - | 26.511 | - |
| Calcium (ppm) | | | | | | | |
| Group (A) | 1 | 45.923 | 45.923 | 56.464 | 0.017* | 0.835 | 91.19 |
| Type (B) | 2 | 1.626 | 0.813 | 4.372 | 0.052* | 0.023 | 2.54 |
| Variety (C) | 8 | 1.488 | 0.186 | 4.497 | 0.000* | 0.016 | 1.75 |
| Error | 96 | 3.971 | 0.041 | - | - | 0.041 | 4.52 |
| Total | 107 | 53.009 | | - | - | 0.916 | - |
| Magnesium (ppm) | | | | | | | |
| Group (A) | 1 | 101.003 | 101.003 | 44.087 | 0.022* | 1.828 | 93.39 |
| Type (B) | 2 | 4.582 | 2.291 | 12.104 | 0.004* | 0.078 | 3.98 |
| Variety (C) | 8 | 1.514 | 0.189 | 5.527 | 0.000* | 0.017 | 0.88 |
| Error | 96 | 3.287 | 0.034 | - | - | 0.034 | 1.75 |
| Total | 107 | 110.386 | - | - | - | 1.957 | - |

**Table 2. Amount of N, P, K, Ca, Mg in reaction groups (inoculated and un-inoculated), types (resistant and susceptible) and in varieties/ cultivars of citrus plant leaves.**

| Nitrogen (%) | | | | | | | | | | | | |
|---|---|---|---|---|---|---|---|---|---|---|---|---|
| Varieties (C) | Mayer lemon | | Eruka lemon | | Citron | | Valentia lnate | | Washington navel | | Kinnow | |
| Type (B) | Resistant | | | | | | Susceptible | | | | | |
| Group (A) | Inoc. | Unin. | Inoc. | Unin. | Inoc. | Unin. | Inoc. | Unin. | Inoc. | Unin. | Inoc. | Unin. |
| Amount of N in (C) | 3.95 | 9.95 | 3.22 | 9.2 | 3.80 | 9.7 | 2.30 | 7.81 | 2.64 | 8.07 | 3.00 | 7.64 |
| Av. Amount of N in (C) | 6.95 | | 6.21 | | 6.75 | | 5.05 | | 5.85 | | 5.32 | |
| Av. Amount of N in (B) | Resistant = 6.63 | | | | | | Susceptible = 6.07 | | | | | |
| Av. Amount of N in (A) | Inoculated = 3.15 | | | | | | Un-Inoculated = 8.56 | | | | | |
| Phosphorus (%) | | | | | | | | | | | | |
| Amount of P in (C) | 0.90 | 2.01 | 0.85 | 1.94 | 0.92 | 2.08 | 0.54 | 1.64 | 0.64 | 1.75 | 0.74 | 1.85 |
| Av. Amount of P in (C) | 1.45 | | 1.39 | | 1.5 | | 1.09 | | 1.19 | | 1.29 | |
| Av. Amount of P in (B) | Resistant = 1.44 | | | | | | Susceptible = 1.19 | | | | | |
| Av. Amount of P in (A) | Inoculated = 0.76 | | | | | | Un-Inoculated = 1.87 | | | | | |
| Potassium (ppm) | | | | | | | | | | | | |
| Amount of K in (C) | 5.17 | 12.22 | 4.42 | 11.81 | 4.82 | 12.01 | 2.36 | 9.17 | 2.77 | 9.73 | 2.45 | 9.51 |
| Av. Amount of K in (C) | 8.69 | | 8.11 | | 8.41 | | 5.76 | | 6.25 | | 5.98 | |
| Av. Amount of K in (B) | Resistant = 8.40 | | | | | | Susceptible = 5.99 | | | | | |
| Av. Amount of K in (A) | Inoculated = 3.66 | | | | | | Un-Inoculated = 10.74 | | | | | |
| Calcium (ppm) | | | | | | | | | | | | |
| Amount of Ca in (C) | 1.63 | 2.68 | 1.44 | 2.88 | 1.53 | 2.93 | 1.1 | 2.49 | 1.22 | 2.6 | 1.52 | 2.69 |
| Av. Amount of Ca in (C) | 2.1 | | 2.16 | | 2.23 | | 1.79 | | 1.91 | | 2.1 | |
| Av. Amount of Ca in (B) | Resistant = 2.1 | | | | | | Susceptible = 1.93 | | | | | |
| Av. Amount of Ca in (A) | Inoculated = 1.40 | | | | | | Un-Inoculated = 2.71 | | | | | |
| Magnesium (ppm) | | | | | | | | | | | | |
| Amount of Mg in (C) | 0.78 | 3.1 | 0.68 | 2.74 | 0.87 | 2.86 | 0.5 | 2.17 | 0.59 | 2.37 | 0.74 | 2.51 |
| Av. Amount of Mg in (C) | 1.94 | | 1.71 | | 1.86 | | 1.33 | | 1.48 | | 1.6 | |
| Av. Amount of Mg in (B) | Resistant = 1.83 | | | | | | Susceptible = 1.47 | | | | | |
| Av. Amount of Mg in (A) | Inoculated = 0.69 | | | | | | Un-Inoculated = 2.62 | | | | | |

respectively (Table 2). Inoculated and un-inoculated leaves of citrus plants exhibited significant variation, with extent of 3.66% and 10.74%, respectively, during the disease stress which accounts for 88.79% of the total variance (Tables 1 and 2). Significant difference was also expressed by the resistant (8.22%) and susceptible (5.99%) plants. Varieties expressed their natural affinities against K contents, expressing 0.29% of the total variance (Table 1). Mayer lemon showed maximum concentration (8.69%) of K while Valentia late exhibited minimum concentration 5.99% of Potassium (Table 2) whereas in calcium, significant difference was recorded among the un-inoculated as well as inoculated leaves of citrus plants with the extent of 2.71 and 1.40ppm respectively under disease stress with 91.19% of total variance as indicated by Table 2. Varieties which were marked as resistant and susceptible also expressed signification alteration and varieties revealed total variance of 2.54% as shown by Table 1. Varieties indicated 1.75% of total variance (Table 1). Maximum concentration 2.23ppm was expressed by Citron (2.23ppm) while Valentia late (1.79ppm) exhibited minimum concentration of Calcium (Table 2).

In case of magnesium concentration, significant alteration was recorded among un-inoculated (2.62 ppm) and inoculated (0.69 ppm) leaves of citrus plants under stress conditions, accounting 93.39% of the total variance as shown by Tables 1 and 2. Plants categorized as resistant and susceptible types also indicated significant alteration, with 1.83 ppm and 1.47 ppm,

**Table 3. Nested structured ANOVA for Zn, Na, Fe and Cu contents of inoculated and un-inoculated citrus plant leaves.**

| SOV | DF | SS | MS | F value | Pr>F | Variance component | % of total variance component |
|---|---|---|---|---|---|---|---|
| **Zinc (ppm)** | | | | | | | |
| Group (A) | 1 | 95.536 | 95.536 | 16.165 | 0.057* | 1.660 | 87.57 |
| Type (B) | 2 | 11.819 | 5.909 | 26.986 | 0.000* | 0.211 | 11.12 |
| Variety (C) | 8 | 1.752 | 0.219 | 341.359 | 0.000* | 0.024 | 1.28 |
| Error | 96 | 0.061 | 0.000 | - | - | 0.001 | 0.03 |
| Total | 107 | 109.169 | - | - | - | 1.895 | - |
| **Sodium (ppm)** | | | | | | | |
| Group (A) | 1 | 49.153 | 49.153 | 30.819 | 0.031* | 0.881 | 92.99 |
| Type (B) | 2 | 3.189 | 1.594 | 18.156 | 0.001* | 0.056 | 5.89 |
| Variety (C) | 8 | 0.702 | 0.087 | 94.683 | 0.000* | 0.010 | 1.02 |
| Error | 96 | 0.089 | 0.000 | - | - | 0.001 | 0.10 |
| Total | 107 | | - | - | - | 0.947 | - |
| **Iron (ppm)** | | | | | | | |
| Group (A) | 1 | 161.428 | 161.428 | 9.144 | 0.054* | 2.662 | 79.18 |
| Type (B) | 2 | 35.307 | 17.653 | 44.512 | 0.000* | 0.639 | 19.01 |
| Variety (C) | 8 | 3.172 | 0.396 | 20.646 | 0.000* | 0.042 | 1.25 |
| Error | 96 | 1.844 | 0.019 | - | - | 0.019 | 0.57 |
| Total | 107 | 201.753 | - | - | - | 3.363 | - |
| **Copper (ppm)** | | | | | | | |
| Group (A) | 1 | 0.1234 | 0.1234 | 19.956 | 0.047* | 0.002 | 88.90 |
| Type (B) | 2 | 0.0124 | 0.0062 | 13.430 | 0.003* | 0.000 | 8.68 |
| Variety (C) | 8 | 0.0037 | 0.0005 | 51.970 | 0.000* | 0.000 | 2.05 |
| Error | 96 | 0.0009 | 0.0000 | - | - | 0.000 | 0.36 |
| Total | 107 | 0.1403 | - | - | - | 0.002 | - |

respectively (Table 2). Varieties with 0.88% of the total variance showed their natural affinities regarding magnesium concentration (Table 1). Mayer lemon (1.94 ppm) and Valentia late (1.33 ppm) indicated their highest and lowest concentrations, consistently (Table 2).

Un-inoculated (2.30ppm) and inoculated (0.42ppm) leaves of citrus plant expressed significant difference regarding zinc concentration with total variance of 87.57% (Tables 3 and 4). Similar findings were obtained between resistant (2.21ppm) and susceptible (1.09ppm) varieties. Varieties with 1.28% of total variance also showed their natural affinities regarding zinc concentration (Table 3). Concentrations of 3.5 ppm and 0.99 ppm were displayed by Mayer lemon (Table 4) and Valentia late, accounting for 11.12% of the total variance, as shown by (Table 3).

Regarding sodium concentration, significant alterations were recorded across the inoculated (0.74 ppm) and un-inoculated (2.08 ppm) citrus plant leaves respectively with 92.99% of the total variance (Tables 3 & 4). Resistant (1.58 ppm) as well as susceptible (1.24 ppm) varieties also expressed a significant alteration in sodium contents under diseased conditions. Varieties expressed 1.02% of the total variance as shown by Table 3. Mayer lemon with 1.70 ppm and Valentia late with 1.18 ppm showed highest and lowest Na concentration (Table 4).

In case of iron, Substantial alterations were recorded among un-inoculated and inoculated leaves citrus plant (averaging to 3.57 ppm and 1.13 ppm respectively) under diseased condition with 79.18% of the total variance at P<0.05 as shown in Tables 3 and 4. Significant difference was also indicated by resistant (2.89 ppm) and susceptible plants (1.81 ppm) as shown by Table 4. 1.25% of the total variance was indicated by the varieties in their natural affinities

**Table 4. Amount of Zn, Na, Fe, Cu in reaction groups (inoculated and un-inoculated), types (resistant and susceptible) and in varieties/ cultivars of citrus plant leaves.**

| | | | | | | | | | | | | |
|---|---|---|---|---|---|---|---|---|---|---|---|---|
| **Zinc (ppm)** | | | | | | | | | | | | |
| Varieties (C) | Mayer lemon | | Eruka lemon | | Citron | | Valentia lnate | | Washington navel | | Kinnow | |
| Type (B) | **Resistant** | | | | | | **Susceptible** | | | | | |
| Group (A) | Inoc. | Unin. | Inoc. | Unin. | Inoc. | Unin. | Inoc. | Unin. | Inoc. | Unin. | Inoc. | Unin. |
| Amount of Zn in (C) | 0.51 | 2.99 | 0.47 | 2.55 | 0.50 | 2.75 | 0.33 | 1.66 | 0.35 | 1.77 | 0.37 | 2.09 |
| Av. Amount of Zn in (C) | 3.5 | | 1.51 | | 1.62 | | 0.99 | | 1.06 | | 1.23 | |
| Av. Amount of Zn in (B) | Resistant = 2.21 | | | | | | Susceptible = 1.09 | | | | | |
| Av. Amount of Zn in (A) | Inoculated = 0.42 | | | | | | Un-Inoculated = 2.30 | | | | | |
| **Sodium (ppm)** | | | | | | | | | | | | |
| Amount of Na in (C) | 1.02 | 2.39 | 0.76 | 2.2 | 0.85 | 2.28 | 0.56 | 1.80 | 0.68 | 1.97 | 0.57 | 1.89 |
| Av. Amount of Na in (C) | 1.70 | | 1.48 | | 1.56 | | 1.18 | | 1.32 | | 1.23 | |
| Av. Amount of Na in (B) | Resistant = 1.58 | | | | | | Susceptible = 1.24 | | | | | |
| Av. Amount of Na in (A) | Inoculated = 0.74 | | | | | | Un-Inoculated = 2.08 | | | | | |
| **Iron (ppm)** | | | | | | | | | | | | |
| Amount of Fe in (C) | 1.63 | 4.46 | 1.31 | 4.14 | 1.50 | 4.32 | 0.63 | 2.74 | 0.80 | 2.59 | 0.93 | 3.21 |
| Av. Amount of Fe in (C) | 3.04 | | 2.72 | | 2.91 | | 1.68 | | 1.69 | | 2.07 | |
| Av. Amount of Fe in (B) | Resistant = 2.89 | | | | | | Susceptible = 1.81 | | | | | |
| Av. Amount of Fe in (A) | Inoculated = 1.13 | | | | | | Un-Inoculated = 3.57 | | | | | |
| **Copper (ppm)** | | | | | | | | | | | | |
| Amount of Cu in (C) | 0.15 | 0.22 | 0.15 | 0.22 | 0.14 | 0.20 | 0.13 | 0.19 | 0.12 | 0.19 | 0.12 | 0.19 |
| Av. Amount of Cu in (C) | 0.37 | | 0.37 | | 0.34 | | 0.32 | | 0.31 | | 0.31 | |
| Av. Amount of Cu in (B) | Resistant = 0.36 | | | | | | Susceptible = 0.31 | | | | | |
| Av. Amount of Cu in (A) | Inoculated = 0.13 | | | | | | Un-Inoculated = 0.20 | | | | | |

towards Fe concentration (Table 3). Maximum concentration was expressed by the variety "Mayer lemon" reaching 3.04 ppm, while the minimum was observed in "Valentia late" at 1.68 ppm as indicated by Table 4. Inoculated (0.13 ppm) and un-inoculated (0.20 ppm) leaves of citrus indicated significant variation in Cu, accounting for 88.90% of total variance (Tables 3 and 4).Similar results were observed between resistant (0.36 ppm) and susceptible (0.31 ppm) plants, while varieties with 2.05% of the total variance showed their natural affinities regarding copper concentrations. Contents of 0.37 ppm and 0.32 ppm were retained by Mayer lemon and Valentia late, accounting for 8.68% of the total variance (Table 4).

## Discussion

Mineral contents have a key role in plant disease interaction. Minerals have a distinct effect on the vigor, physiology, resistance and biochemical reactions occurring within the plants. Pathogen attack on the plant to snatch minerals for performing their activities. Excess or deficiency of these minerals may enhance resistance or susceptibility of the plants or enhance/ reduce the aggressiveness of the pathogen. Appropriate provision of the nutrients to the host plants can reduce the severity of disease by inducing resistance in host plants towards different microbes. All biochemical, physiological and metabolic activities were observed to be more efficient than those plants which received excess or deficient amount of nutrients [26]. The current study was planned to determine the alterations in mineral contents of susceptible and resistant citrus cultivars as a result of attack of brown spot in citrus. In the current study, significant alterations in conc. of N, K, P, Ca, Mg, Cu, Fe, Zn, and Na were noticed. These findings are helpful for researchers and scientists for devised a solid solution for brown spot in citrus.

Nitrogen plays a key role in normal physiological functioning of plants, as it is a major component of proteins necessary for plant growth. It is also involved in the manufacturing of many living cells [27]. Excess or deficiency of nitrogen increases the severity or incidence of diseases [28]. After attack, pathogen use nitrogen resources from colonized tissues and leaves. In general, nitrogen is required to provide plants with amino acids necessary for their growth and resistance to various pathogens, while plants deficient in N become weaker and grow slowly. Such plants are more susceptible to pathogen attacks, as indicated by Agrios, [29]. Increased synthesis of nitrogen-carrying compounds and amino acids, like hydroxyproline, is essential for active plant defense. Glycoproteins deposited in the cell wall contribute to resistance by trapping the pathogens [30]. However, pathogen interfere with nitrogen-mediated defenses, such as antimicrobial proteins, phytoalexins, amino acids, organic acid, as well as defense-related enzymes [31, 32]. However, the accurate provision of nitrogen to plants increases the availability of nitrogenous compounds to defend against pathogens [33]. In current findings, N (%) was significantly high in resistant than susceptible and in un-inoculated than inoculated leaves of citrus plants. This is also supported by Veverka et al. and Ali et al. [34, 35], who described that N contents are reduced after the attack of fungi.

Phosphorus is constituent of energy-containing molecules like ATP, DNA, RNA, and membrane phospholipids, which are involved in the defense system of plants. It also plays a basic role in cell division as well as growth of meristematic tissues. Phosphorus is required for regulation of enzyme to perform reactions and cell metabolic pathways [36] while reduced availability of phosphorus to plants affects synthesis of ATP and phospholipids thereby making the plants susceptible towards diseases. However, foliar application of phosphorus offers systemic resistance towards various pathogens [37]. In present study, concentration of phosphorus was maximum in resistant and un-inoculated citrus leaves than susceptible and inoculated. These findings are in agreement with the studies of [35, 38, 39] who describes that amount of phosphorus decreases in plants after attack of Alternaria fungi.

Potassium has key role in growth and physiological functions of plants. It also contributes in many metabolic as well as biochemical reactions occurred in plant cell [40, 41]. Potassium is involved in synthesis of protein, enzyme activation, movements of stomata and photosynthesis [42] while its deficiency causes the conductance of stomata, enhances the resistance in mesophyll, reduce the carboxylase and ribose 1,5 bisphosphate activaty thereby lowering overall photosynthetic process of plants [43]. Moreover, plants which are deficient in K contents possess thin cell wall, shorter and smaller root and sugar accumulation in leaves which favors the entry and infection of the pathogens [44]. In current study, it was observed that resistant and un-inoculated citrus varieties contained more amount of potassium than susceptible and inoculated varieties of citrus which is also witnessed by [35, 45, 46] who reported that plants which are deficient in K contents are more vulnerable to attack of diseases.

Plant take up calcium in the form of Ca+ which is primarily used in growth of leaves, roots as well as in the absorption of other essential nutrients. It also plays a key role in maintaining permeability and integrity of cell and is an integral component of middle lamella [46]. Many pathogens including fungi invade pectolytic enzymes like pectate transeliminase and polyglacturonases that may dissolve middle lamella thereby favoring infection while Ca compounds play a key role to constitute healthy and stable cell wall as well as inhibit synthesis of these pectolytic enzymes thus reducing chances of infection [47]. In present study, reduction in concentration of Ca was observed in infected citrus leaves which were witnessed by the findings of [48].

Magnesium is required for preservation of ribosomal structure which is associated with high level of protein, active mitosis, metabolism of carbohydrates and for oxidative phosphorylation. It plays role in energy transfer reactions in plants, DNA and RNA formation, respiration and act as cofactor for several enzymes in the cell [49]. Plant performed physical and

chemical defenses and active defenses are produced after infection [50] both of which needs energy from photosynthesis involving magnesium as a part of chlorophyll. Mg is also a constituent of middle lamella but its deficiency during growth reduces the integrity of middle lamella as well as production of energy that is required for defense mechanism thereby making the plants susceptible to diseases. Moreover, accumulation of sucrose and starch in leaf tissues under magnesium deficiency may provide a nutrient-rich environment conducive for the growth and pathogen attack. In present study, a significant reduction in concentration of magnesium was determined in susceptible and inoculated citrus varieties than resistant and uninoculated cultivars as indicated by the [35] who reported that Mg concentration was decreased in plants infected with fungal pathogens.

Zinc is an important constituent of enzyme systems that is involved in regulation of metabolic processes in many plant such as superoxide dismutase, RNA polymerases, alcohol dehydrogenase and protein synthesis [13]. Zinc plays a vital role in formation of auxin, chlorophyll synthesis as well as improving uptake of water by plants [51] while its deficiency reduce plant growth and make them susceptible against various abiotic and biotic factors [52]. A significant difference in Znic contents was observed in the citrus cultivars where less amount of Zn in inoculated and susceptible citrus cultivars than resistant and un-inoculated cultivars was observed which is also witnessed by the studies of [38] who observed reduction in Zn contents due to attack of diseases.

Fe has prime role in production of chlorophyll, protein synthesis and nucleic acid metabolism. It also triggers the enzymes activation that play a role in host defense system while its deficiency results in chlorosis of young leaves thereby reducing the photosynthetic efficiency of plants [48]. Significant changes were observed in iron contents of citrus cultivars. Inoculated and susceptible citrus varieties contain less amount of iron as compared to un-inoculated and resistant cultivars which is also supported by the findings of [38, 53] who observed higher amount iron in resistant plants after attack of pathogens.

Cu plays a vital role in protein and carbohydrate metabolism is a main constituent of lignin. It is also involved in photosynthesis, chlorophyll formation, enzyme systems, oxidation-reduction reactions as well as root metabolism. All these processes are influenced by the deficiency of copper in host plants thereby making the plants more susceptible to pathogen attack [54]. In current study, higher amount of copper was determined in resistant and un-inoculated citrus cultivars than susceptible and inoculated varieties. The results of current study are supported by the findings of [53] who described that higher amounts of iron and copper were present in resistant plants. It was confirmed from the outcomes of present work that there is direct relationship between reduction in amount of nutrients and disease occurrence as pathogen required nutrition for its growth and multiplication. Similar outcomes were also described by [13, 38] who described the interaction of nutrients and disease severity. Future research should prioritize examining specific nutrient supplements to boost citrus resilience against brown spot stress and investigating the role of soil microbiota in nutrient uptake and overall plant health. It is also crucial to conduct comparative studies on the biochemical responses of various citrus varieties to stress, alongside metabolomic profiling to identify biomarkers of stress tolerance.

## Conclusion

All resistant and un-inoculated citrus varieties contained higher amounts of nitrogen, phosphorus, potassium, sodium, calcium, magnesium, copper, zinc and iron as compared to susceptible and inoculated ones. So, accurate provision of these nutrients is helpful in improving the physical as well as biochemical processes of citrus plants and also a most promising strategy to manage brown spot of citrus.

## Supporting information

**S1 Data.**
(XLSX)

## Acknowledgments

I would like to pay gratitude to Citrus Pathology Laboratory. Department of Plant Pathology, University of Agriculture, Faisalabad, Pakistan who provide me research facilities and technical assistance under NRPU-HEC # 9315.

## Author Contributions

**Formal analysis:** Nadeem Akbar.

**Investigation:** Shahid Iqbal.

**Project administration:** Muhammad Atiq.

**Supervision:** Muhammad Atiq.

**Visualization:** Nadeem Akbar.

**Writing – original draft:** Shahid Iqbal.

**Writing – review & editing:** Shahbaz Talib Sahi, Nasir Ahmed Rajput.

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
