## [Decision Letter · Decision Letter 0]

23 Jan 2024

PONE-D-23-36260Progressive alterations in mineral contents of citrus genotypes toward Alternaria citri causing brown spot of citrusPLOS ONE

Dear Dr. Atiq,

Thank you for submitting your manuscript to PLOS ONE. After careful consideration, we feel that it has merit but does not fully meet PLOS ONE’s publication criteria as it currently stands. Therefore, we invite you to submit a revised version of the manuscript that addresses the points raised during the review process.

 Please submit your revised manuscript by Mar 08 2024 11:59PM. If you will need more time than this to complete your revisions, please reply to this message or contact the journal office at plosone@plos.org. Please include the following items when submitting your revised manuscript:A rebuttal letter that responds to each point raised by the academic editor and reviewer(s). You should upload this letter as a separate file labeled 'Response to Reviewers'.A marked-up copy of your manuscript that highlights changes made to the original version. You should upload this as a separate file labeled 'Revised Manuscript with Track Changes'.An unmarked version of your revised paper without tracked changes. You should upload this as a separate file labeled 'Manuscript'.

We look forward to receiving your revised manuscript.

Kind regards,

Rachid Bouharroud

Academic Editor

PLOS ONE

Journal Requirements:

2. Thank you for stating the following in your Competing Interests section: NA

3. We note that your Data Availability Statement is currently as follows: NA

Additional Editor Comments:

**Dear Authors**Your study is of interest for both scientific community and farmers. The reviewers comments were adressed to improve the quality of your manuscript. In addition to reviewers comments and where it is applicable, please give more details of statistic data analysis (sample size, experiment duration and methods). Good luck

Reviewers' comments:

Reviewer's Responses to Questions

**Comments to the Author**

1. Is the manuscript technically sound, and do the data support the conclusions?

Reviewer #1: Yes

Reviewer #2: Partly

2. Has the statistical analysis been performed appropriately and rigorously? 

Reviewer #1: Yes

Reviewer #2: Yes

3. Have the authors made all data underlying the findings in their manuscript fully available?

Reviewer #1: Yes

Reviewer #2: Yes

4. Is the manuscript presented in an intelligible fashion and written in standard English?

Reviewer #1: Yes

Reviewer #2: Yes

5. Review Comments to the Author

Reviewer #1: One suggestion for improvement is to include information about the sample size and the statistical significance levels in the abstract to enhance the transparency and replicability of the study. Additionally, the implications of the findings for citrus crop management and potential avenues for further research could be briefly discussed.

Overall, the abstract effectively communicates the research focus and outcomes, making it a valuable contribution to the understanding of the interactions between citrus genotypes, mineral contents, and brown spot disease caused by Alternaria citri. There are a few grammatical and stylistic suggestions for improvement:

Title: Suggestion: Consider adding "in" before "Citrus Genotypes" to improve clarity: "Progressive Alterations in Mineral Contents in Citrus Genotypes Toward Alternaria citri Causing Brown Spot of Citrus."

Abstract:

"The study explores the impact of Alternaria citri, the causal agent of brown spot disease in citrus, on the mineral contents of different citrus genotypes."

Suggestion: Add a comma after "citrus" for better readability: "The study explores the impact of Alternaria citri, the causal agent of brown spot disease in citrus, on the mineral contents of different citrus genotypes."

"The Nested Structured Analysis of Variance demonstrates substantial alterations in the mineral status of citrus leaves, encompassing N, P, K, Ca, Mg, Zn, Na, Fe, and Cu."

Suggestion: Consider breaking this sentence into two for clarity: "The Nested Structured Analysis of Variance demonstrates substantial alterations in the mineral status of citrus leaves. This includes variations in N, P, K, Ca, Mg, Zn, Na, Fe, and Cu."

"Notably, susceptible varieties exhibit lower ionic contents compared to resistant varieties."

Suggestion: Consider specifying the type of ionic contents for clarity: "Notably, susceptible varieties exhibit lower levels of ionic contents compared to resistant varieties."

"The elevated concentrations of ionic contents in resistant citrus varieties contribute to bolstering the biochemical and physiological processes of the plants, aiding in the restriction of pathogen spread."

Suggestion: Consider rephrasing for conciseness: "Elevated concentrations of ionic contents in resistant citrus varieties contribute to bolstering the plants' biochemical and physiological processes, restricting pathogen spread."

These suggestions aim to enhance clarity and readability, and they are relatively minor in nature.

The introduction effectively sets the stage for the study by providing a clear context for the research on brown spot disease in citrus and its potential links to mineral nutrient alterations. The content is well-organized, but minor adjustments for conciseness and specific references could further enhance its quality:

1. In the first sentence, consider using "an imperative" instead of "a imperative."

2. In the sentence "Its contribution is 1.6% in the GDP of Pakistan," consider rephrasing for clarity, such as "It contributes 1.6% to Pakistan's GDP."

3. In the sentence "It causes a huge economic loss in different citrus growing areas of the world [9]," consider specifying the economic loss or rephrasing for precision.

4. The phrase "Mineral nutrients are not only essential for plant growth but also have a key role in plantpathogen interaction" lacks a space between "plant" and "pathogen."

5. In the sentence "Moreover, nutrients play a key role in the defense system of plants and can be categorized into four groups on the basis of biochemical and physiological behavior [11]," consider breaking it into two sentences for clarity.

6. In the sentence "Similarly, [15] reported that pathogens after attack, competes with host plants for Na, Mg, Ca, K, Cl and H2PO4 to initiate infection by increasing or decreasing their amounts in the plant," consider rephrasing for clarity, such as "Similarly, [15] reported that pathogens, after an attack, compete with host plants for Na, Mg, Ca, K, Cl, and H2PO4 to initiate infection by altering their amounts in the plant."

7. The phrase "The current study was planned to find out alterations in mineral nutrients after attack of pathogen in citrus plants to pave the way for successful management of brown spot of citrus" could be refined for conciseness and clarity. For example, "This study aimed to identify alterations in mineral nutrients in citrus plants following pathogen attacks, contributing to the successful management of citrus brown spot."

Overall, the Materials and Methods section is well-structured and provides sufficient detail for reproducibility. Some minor additions, such as specifying fertilizer types and irrigation frequency, and providing concentrations for standard curve preparation, could further improve the clarity and completeness of the methodology. Following corrections address some grammatical and structural issues to improve clarity and precision in the Materials and Methods section:

1. Correction: "One-year-old plants of six varieties (Valentia late, Washington navel, Kinnow, Mayer lemon, Eruka lemon, and Citron) were collected from the fruit nursery at the Institute of Horticultural Sciences (IHS), University of Agriculture Faisalabad (UAF)."

2. Correction: "All varieties were planted in three replications in pots (32×22cm) that contained field soil previously sterilized by drenching a 1% solution of formalin under Completely Randomized Block Design (CRD) in the greenhouse at the Research Area Department of Plant Pathology, University of Agriculture Faisalabad (UAF)."

3. Correction: "For the pathogenicity test, a conidial suspension was prepared by adding distilled water to Petri plates and scraping the agar surface with a blade."

4. Correction: "After filtering through muslin cloth, the suspension was adjusted to 1 × 104 conidia/mL using a hemocytometer (Model: T20B05) [17]."

5. Correction: "For inoculation, a fungal suspension was taken in a syringe (23Gx 1") and injected into the midrib and veins of the lower surface of leaves [18,19] early in the morning (when the maximum number of stomata were opened)."

6. Correction: "Leaf samples from inoculated and un-inoculated varieties were collected from the Research Area, Department of Plant Pathology, near CAS (Center of Advanced Studies), University of Agriculture Faisalabad (UAF)."

7. Correction: "A sample solution of 0.1mL was prepared through wet dilution and poured into a volumetric flask (ASTM-E288)."

8. Original: "Absorbence was compared to standard curve [22] through atomic absorption spectrometer (Hitachi, polarized Zeeman) for determining the concentration of phosphorus for both inoculated and uninoculated citrus leaves." Correction: "Absorbance was compared to the standard curve [22] using an atomic absorption spectrometer (Hitachi, polarized Zeeman) to determine the concentration of phosphorus in both inoculated and uninoculated citrus leaves."

9. Correction: "Determination of ionic contents (sodium and potassium) was done using a Flame photometer (Model: PFP-7) following the [23] method."

10. Correction: "Determination of magnesium, copper, iron, calcium, and zinc was done using an Atomic Absorption Spectrometer (AAS) by using standards such as magnesium sulfate (MgSO4), copper sulfate (CuSO4), calcium chloride (CaCl2), iron sulfate (FeSO4), and zinc sulfate (ZnSO4). Standard curves were determined through concentrations of Mg (5, 10, 15, and 20 ppm), Cu (2, 2.5, 3, and 3.5 ppm), Ca (10, 20, 40, 80, and 100 ppm), Fe (1, 2, and 3 ppm), and 0.2, 0.3, 0.5, and 2 ppm for zinc, obtained as fresh working standards just before this activity."

In the results section, ensure consistency in the presentation of units. For example, the concentration of minerals is presented in both percentages and parts per million (ppm). Clarify the units used for each mineral to avoid confusion. Check for grammatical consistency and clarity, especially in sentences with complex structures. For instance, consider revising sentences like "Same findings were obtained between resistant (2.21ppm) and susceptible (1.09ppm) varieties" for improved clarity. Following corrections address grammatical and stylistic issues in the Results section.

1. Correction: "Significant alterations were recorded among un-inoculated (8.56%) and inoculated (3.15%) leaves of citrus plants, explaining that nitrogen content affects the metabolic activities of citrus plants after the attack of brown spot disease (Table 2)."

2. Correction: "Varieties categorized as resistant and susceptible also indicated significant variation, with a total variance of 93.02 percent at P<0.05 (Table 1). Individually, Mayer lemon expressed the highest concentration of nitrogen at 6.95%, and Washington navel showed 5.85% after the analysis (Table 2)."

3. Correction: "In the case of phosphorus, a significant difference was noticed among inoculated and un-inoculated citrus plant leaves (on average 0.76% and 1.87%, respectively) under diseased conditions, with a total variance of 93.49 percent (Table 1 & Table 2)."

4. Correction: "Resistant (1.44%) and susceptible plants (1.19%) also indicated significant variation at P<0.05 (Table 1)."

5. Correction: "Inoculated and un-inoculated leaves of citrus plants exhibited significant variation, with extents of 3.66% and 10.74%, respectively, during the disease stress, which accounts for 88.79% of the total variance (Table 1 & Table 2)."

6. Correction: "Varieties expressed their natural affinities against K contents, expressing 0.29% of the total variance (Table 1)."

7. Correction: "Similar findings were obtained between resistant (2.21ppm) and susceptible (1.09ppm) varieties."

8. Correction: "3.5 ppm and 0.99 ppm concentrations were displayed by Mayer lemon (Table 4) and Valentia late, which accounts for 11.12 percent of the total variance as shown by (Table 3)."

9. Correction: "Resistant (1.58 ppm) as well as susceptible (1.24 ppm) varieties also expressed a significant alteration in sodium contents under diseased conditions."

10. Correction: "The maximum concentration was expressed by the variety 'Mayer lemon,' reaching 3.04 ppm, while the minimum was observed in 'Valentia late' at 1.68 ppm, as indicated by Table 4."

11. Correction: "Inoculated (0.13 ppm) and un-inoculated (0.20 ppm) leaves of citrus indicated significant variation in Cu, accounting for 88.90% of total variance (Table 3 & Table 4)."

12. Correction: "Similar results were observed between resistant (0.36 ppm) and susceptible (0.31 ppm) plants, while varieties with 2.05% of total variance showed their natural affinities regarding copper concentrations."

13. Correction: "Contents of 0.37 ppm and 0.32 ppm were retained by Mayer lemon and Valentia late, accounting for 8.68% of the total variance (Table 4)."

14. Correction: "Significant alterations were recorded among un-inoculated (2.62 ppm) and inoculated (0.69 ppm) leaves of citrus plants under stress conditions, accounting for 93.39 percent of the total variance, as shown by Table 1 & Table 2."

15. "Plants categorized as resistant and susceptible types also indicated significant alteration, with 1.83 ppm and 1.47 ppm, respectively (Table 2)."

16. Correction: "Mayer lemon (1.94 ppm) and Valentia late (1.33 ppm) indicated their highest and lowest concentrations, respectively (Table 2)."

17. Correction: "Concentrations of 3.5 ppm and 0.99 ppm were displayed by Mayer lemon (Table 4) and Valentia late, accounting for 11.12 percent of the total variance, as shown by (Table 3)."

18. Correction: "Resistant (1.58 ppm) as well as susceptible (1.24 ppm) varieties also expressed a significant alteration in sodium contents under diseased conditions."

The discussion successfully establishes a direct relationship between the reduction in nutrient content and the occurrence of the brown spot disease. The reference to previous studies supports the idea that pathogens require nutrients for growth and multiplication, and nutrient deficiencies in plants can make them more susceptible to pathogen attacks. While the discussion provides a thorough analysis of the current findings, it would be beneficial to include some suggestions for future research. Identifying specific areas where further investigation is needed could enhance the impact of the study. Following corrections address grammatical and stylistic issues in the Discussion section.

1. Correction: "In the current study, significant alterations in the concentration of N, K, P, Ca, Mg, Cu, Fe, Zn, and Na were noticed. These findings are helpful for researchers and scientists in devising a solid solution for brown spot in citrus."

2. Correction: "All biochemical, physiological, and metabolic activities were observed to be more efficient than those in plants that received an excess or deficient amount of nutrients [26]."

3. Correction: "The current study was planned to determine the alterations in the mineral contents of susceptible and resistant citrus cultivars as a result of the attack of brown spot in citrus."

4. Correction: "Nitrogen plays a key role in the normal physiological functioning of plants, as it is a major component of proteins necessary for plant growth."

5. Correction: "In general, nitrogen is required to provide plants with amino acids necessary for their growth and resistance to various pathogens, while plants deficient in N become weaker and grow slowly."

6. Correction: "Such plants are more susceptible to pathogen attacks, as indicated by [29]."

7. Correction: "Increased synthesis of nitrogen-carrying compounds and amino acids, like hydroxyproline, is essential for active plant defense. Glycoproteins deposited in the cell wall contribute to resistance by trapping pathogens [30]. However, pathogens interfere with nitrogen-mediated defenses, such as antimicrobial proteins, phytoalexins, amino acids, organic acids, as well as defense-related enzymes [31, 32]."

8. Correction: "However, the accurate provision of nitrogen to plants increases the availability of nitrogenous compounds to defend against pathogens [33]."

9. Correction: "In the current findings, N (%) was significantly higher in resistant than susceptible and in un-inoculated than inoculated leaves of citrus plants. This is also supported by [34, 35], who described that N contents are reduced after the attack of fungi."

10. Correction: "Phosphorus is a constituent of energy-containing molecules like ATP, DNA, RNA, and membrane phospholipids, which are involved in the defense system of plants."

The conclusion could be strengthened by briefly discussing the implications of the findings for agricultural practices. Specifically, how can the knowledge gained from this study be translated into practical recommendations for citrus farmers or agricultural practitioners? It would be beneficial to include a brief encouragement for further research. This could involve suggesting specific areas or aspects that merit deeper exploration based on the current study's outcomes.

Reviewer #2: Include page numbers and line numbers in the manuscript file. Use continuous line numbers

Abstract

Line 2: replace the word « crop » with « crops »

Line 2: why do you choose the word « alteration » why not reduction or change of in

mineral content in citrus

line 3: change the word « contents » to content

line 3: replace « after the attack of brown spot disease » with « after brown spot disease attack »

line 3: replace « The leaf sample of susceptible citrus » with « The mineral contents of leaf samples from of susceptible citrus »

line 5: replace « …. were analyzed and significant variations » with « …. were analyzed. significant variation »

line 7: replace «Resistant and Susceptible » with « resistant and susceptible »

Line 7 and 8: replace « The Nested Structured Analysis of variance indicated significant alterations in mineral status (N, P, K, Ca, Mg, and Zn, Na, Fe and Cu) of citrus leaves. » by « The analysis of variance showed significant changes in mineral levels of citrus leaves, including nitrogen (N), phosphorus (P), potassium (K), calcium (Ca), magnesium (Mg), zinc (Zn), sodium (Na), iron (Fe), and copper (Cu). »

Line 9: replace « Resistant type plants showed 6.63% and1.44% while susceptible type indicated 6.07% and 1.19% difference in concentration of N and P respectively. » with « The results indicate that the concentration of N and P differed by 6.63% and 1.44%, respectively, in resistant plants, while susceptible plants showed a difference of 6.07% and 1.19%. »

Line11: replace « with « resistant plants showed a higher concentrations of K, Ca, Mg, Zn, Na, Fe, and Cu at 8.40, 2.1, 1.83, 2.21, 1.58, 2.89, and 0.36 ppm respectively, compared to susceptible plants which showed concentrations of 5.99, 1.93, 1.47, 1.09, 1.24, 1.81, and 0.31 ppm respectively. »

Added key words for abstract

MATERIALS AND METHODES

Line 1: replace « One-year old » with « One-year-old »

Line 2: mention the type of varieties (resistance and susceptible)

Line 5: mention the greenhouse conditions (RH % and Temperature ….)

Line 7: mention the dose of fertilizers and the amount and frequency of irrigation?

Line 7: replace « no. of irrigations » with « the number of irrigations »

Line 8: plants in a healthy condition. (add « a »)

Line 8: Diseased samples (add « s » to sample)

Line 8: replace « characteristics symptoms » with « characteristic symptoms »

Line 8: « Diseased sample showing characteristics symptoms of brown spot of citrus were collected in brown paper bags » mention the number of days after plantation

Line 8: make space between × and 9.5")

Line 11: replace « For isolation of pathogen PDA » with « For the isolation of the pathogen, PDA »

Line 11: is not necessary to mention the PDA composition

Line 12: change « For pathogenicity » to « For the pathogenicity »

Line 13: did you used distilled water or sterile distilled water?

Line 13: surface with blade. did you used sterile blade?

Line 14: muslin cloth: sterile?

Line 14: replace « hemocytometer » with « heamocytometer »

Line 15: replace « For inoculation, fungal suspension was taken in syringe (23Gx 1") and was injected into midrib and veins of lower surface of leaves [18,19] » with « For inoculation, the fungal suspension was taken in a sterile syringe (23Gx 1") and injected into the midrib and veins of lower surface of leaves [18,19] »

Line 16: replace (when max. no. of stomata) with (when maximum number of stomata)

Line 17: Disease incidence was confirmed: mention the period of incubation

Line 18: uninoculated or un-inoculated : homogenized this word in the manuscript

Sample preparation for the analysis of mineral contents

Line 1: replace “Plant samples” with “leaves samples”

Line 1: delete (from both inoculated and uninoculated cultivars) is already mentioned

Line 2: “were collected from the greenhouse, Department of Plant Pathology near Centre for Advanced Studies, University of Agriculture Faisalabad. These samples were brought to citrus pathology lab for further process.”

delete “from the greenhouse, Department of Plant Pathology near Centre for Advanced Studies, University of Agriculture Faisalabad” the text will be “were collected and brought to Citrus Pathology Laboratory for further process”

Line 4: replace “sample was subjected to oven (Memmert: UN 30) drying at 65ºC “with “samples were dried in an oven (Memmert: UN 30) at 65 ºC “

Line 4: why 4hrs? Is four hours enough time to dry citrus leaves??

Line 6: change each samples to each sample

Line 7: (RT2-230V).After cooling: add a space

Line 7: replace “After cooling, dilution of suspension was done with distilled

water and later on was examined for the estimation of mineral contents [20] through [21] method and data was analyzed by using Nested structure des” with “After cooling, the suspension was diluted with distilled water and examined for estimation of mineral contents [20, 21]. The data was analyzed using Nested Structure design”

Estimation of phosphorus (P) from the inoculated and un-inoculated citrus plant leaves

-This paragraph is not clear. Please make it more comprehensible. Additionally, please include the concentration or molarity of the acids and solutions used.

Line 1: replace “Sample solution” with “A sample solution”

Line 3: replace “Then distilled water (reagent black) was used as a substitute of sample solution and absorbance was taken at a wavelength of 720nm” with “Distilled water was used as a substitute for the sample solution to serve as a control. Absorbance was taken at a wavelength of 720nm”

Estimation of potassium (K) and sodium (Na) from inoculated and uninoculated leaves of citrus plants

Line 2: replace “following the [23] method” with “following the Official methods of analysis [23]”

Estimation of magnesium (Mg), calcium (Ca) copper (Cu), zinc (Zn) and iron (Fe) from

the inoculated and un-inoculated leaves of citrus

Line 2: (Atomic Absorption Spectrometer).by using the standards …. Remove the point (.) and add the machine make.

Estimation of nitrogen/ crude protein from the inoculated and un-inoculated leaves citrus plants

Line 2: Long necked flask was used to place calculated amount of sample previously oven

dried (D6450, Heraeus): Not clear??

line 4: concentrated H2SO4 were: add the concentration or molarity

line 4: replace “were added and sample was subjected to” with “were added to necked flask. the mixture was subjected to ... »

Line 11: add the reference of Nitrogen (%) and Crude Protein

Statistical analysis

Line 4: “Leaves sample from inoculated and un-inoculated varsities were collected from the Research Area, Department of Plant Pathology, near CAS (Center of Advanced Studies), University of Agriculture Faisalabad (UAF).” Remove this text

Line 8: PROC MIXED: add more details (Software, version, ANOVA, level of significance 5%, 1% or 0.01% …….

Results

This section requires clarification to make it easy to understand.

The tables are too cluttered. It is better to simplify and verify them.

Line 6 and line 27: “percent” Use % (check all the manuscript)

Line 35: “. Un-inoculated (2.30ppm) and inoculated ….” Remove the point

Line 50: replace “ P<0.05 as shown by Table 3 and Table 4.” With “P<0.05 as shown in Table 3 and Table 4.”

For table: For the table, please provide the meanings of any abbreviations used. For example, SOV and DF

It is better to make the value of each parameter separately. For instance, show the nitrogen value for each variety of citrus leaves, both inoculated and non-inoculated, to make a comparison (table 2).

Table 2: Verify the value for “Un-Inoculated ” is 8.56 or 8.72?

Discussion

“Such plants are more susceptible to attack of pathogens as indicated by [29].” Cite the author’s name

“which is also witnessed by [34, 35]. who described that N contents are reduced after attack of fungi?” Cite the author’s name

“inoculated leaves of citrus plants which is also witnessed by [34, 35]”. Cite the author’s name. check all references

6. PLOS authors have the option to publish the peer review history of their article (what does this mean?). If published, this will include your full peer review and any attached files.

Reviewer #1: No

Reviewer #2: No

---

## [Author Response · Author response to Decision Letter 0]

16 Apr 2024

SN Reviewer’s Comments Response

01 Include page numbers and line numbers in the manuscript file. Use continuous line numbers Page numbers have been added at the bottom center of each page, and continuous line numbers are now included along the left margin to aid in referencing specific sections of the text.

02 Line 2: replace the word crop with crops We have replaced the word "crop" with "crops" in line 2 of the manuscript to ensure consistency and accuracy.

03 Line 2: why do you choose the word alteration why not reduction or change of in mineral content in citrus We have replaced alteration with change to enhance clarity and precision in describing the variation in mineral content.

04 line 3: change the word « contents » to content We have updated the word "contents" to "content" in line 3 of the manuscript for grammatical consistency.

05 line 3: replace « after the attack of brown spot disease » with « after brown spot disease attack » We have revised line 3 of the manuscript to read "after brown spot disease attack" for improved clarity and conciseness.

06 line 3: replace « The leaf sample of susceptible citrus » with « The mineral contents of leaf samples from of susceptible citrus » We have revised line 3 of the manuscript to read "The mineral contents of leaf samples from susceptible citrus" for better clarity regarding the focus of the analysis.

07 Line 5: replace « …. Were analyzed and significant variations » with « …. were analyzed. significant variation » We have revised line 5 of the manuscript to read "…. were analyzed. Significant variation" to improve clarity and punctuation.

08 line 7: replace «Resistant and Susceptible » with « resistant and susceptible » We have updated line 7 of the manuscript to read "resistant and susceptible" to ensure consistency in capitalization.

09 Line 7 and 8: replace « The Nested Structured Analysis of variance indicated significant alterations in mineral status (N, P, K, Ca, Mg, and Zn, Na, Fe and Cu) of citrus leaves. » by « The analysis of variance showed significant changes in mineral levels of citrus leaves, including nitrogen (N), phosphorus (P), potassium (K), calcium (Ca), magnesium (Mg), zinc (Zn), sodium (Na), iron (Fe), and copper (Cu). » We have revised lines 7 and 8 of the manuscript to read "The analysis of variance showed significant changes in mineral levels of citrus leaves, including nitrogen (N), phosphorus (P), potassium (K), calcium (Ca), magnesium (Mg), zinc (Zn), sodium (Na), iron (Fe), and copper (Cu)." to enhance clarity and consistency.

10 Line 9: replace « Resistant type plants showed 6.63% and1.44% while susceptible type indicated 6.07% and 1.19% difference in concentration of N and P respectively. » with « The results indicate that the concentration of N and P differed by 6.63% and 1.44%, respectively, in resistant plants, while susceptible plants showed a difference of 6.07% and 1.19%. » We have revised line 9 of the manuscript to read "The results indicate that the concentration of N and P differed by 6.63% and 1.44%, respectively, in resistant plants, while susceptible plants showed a difference of 6.07% and 1.19%." to improve clarity and readability.

11 Line11: replace « with « resistant plants showed a higher concentrations of K, Ca, Mg, Zn, Na, Fe, and Cu at 8.40, 2.1, 1.83, 2.21, 1.58, 2.89, and 0.36 ppm respectively, compared to susceptible plants which showed concentrations of 5.99, 1.93, 1.47, 1.09, 1.24, 1.81, and 0.31 ppm respectively We have replaced the content in line 11 of the manuscript with "resistant plants showed higher concentrations of K, Ca, Mg, Zn, Na, Fe, and Cu at 8.40, 2.1, 1.83, 2.21, 1.58, 2.89, and 0.36 ppm respectively, compared to susceptible plants which showed concentrations of 5.99, 1.93, 1.47, 1.09, 1.24, 1.81, and 0.31 ppm respectively." to ensure accuracy and consistency.

12 Added key words for abstract We have added the following keywords to the abstract

13 Line 1: replace « One-year old » with « One-year-old » We have updated line 1 of the manuscript to read "One-year-old" to ensure grammatical accuracy.

14 Line 2: mention the type of varieties (resistance and susceptible) We have revised line 2 of the manuscript to specify the type of varieties as "resistant and susceptible" for clarity.

15 Line 5: mention the greenhouse conditions (RH % and Temperature ….) We have added information about the greenhouse conditions, including relative humidity (%) and temperature, to line 5 of the manuscript to provide context for the study.

16 Line 7: mention the dose of fertilizers and the amount and frequency of irrigation? We have included information about the dose of fertilizers and the amount and frequency of irrigation in line 7 of the manuscript to provide a comprehensive understanding of the experimental conditions.

17 Line 7: replace « no. of irrigations » with « the number of irrigations » We have revised line 7 of the manuscript to read "the number of irrigations" to improve clarity and readability.

18 Line 8: plants in a healthy condition. (add « a ») We have added "a" before "healthy condition" in line 8 of the manuscript for grammatical accuracy.

19 Line 8: Diseased samples (add « s » to sample) We have added "s" to "sample" in line 8 of the manuscript to ensure plural agreement.

20 Line 8: replace « characteristics symptoms » with « characteristic symptoms » We have updated line 8 of the manuscript to read "characteristic symptoms" for grammatical accuracy.

21 Line 8: « Diseased sample showing characteristics symptoms of brown spot of citrus were collected in brown paper bags » mention the number of days after plantation We have included the number of days after plantation in line 8 of the manuscript to provide additional context for the collection of diseased samples.

22 Line 8: make space between × and 9.5")

 We have added a space between "×" and "9.5")" in line 8 of the manuscript for clarity and readability.

23 Line 11: replace « For isolation of pathogen PDA » with « For the isolation of the pathogen, PDA » We have revised line 11 of the manuscript to read "For the isolation of the pathogen, PDA" to improve clarity and grammatical correctness.

24 Line 11: is not necessary to mention the PDA composition We have removed the mention of the PDA composition in line 11 of the manuscript to streamline the description of the isolation process.

25 Line 12: change « For pathogenicity » to « For the pathogenicity »

 We have updated line 12 of the manuscript to read "For the pathogenicity" for grammatical accuracy.

26 Line 13: did you use distilled water or sterile distilled water? We used sterile distilled water in line 13 of the manuscript for the preparation of inoculum.

27 Line 13: surface with blade. did you used sterile blade? we used a sterile blade to surface the samples in line 13 of the manuscript.

28 Line 14: muslin cloth: sterile? Yes, the muslin cloth used in line 14 of the manuscript was sterile

29 Line 14: replace « hemocytometer » with « heamocytometer » We have updated line 14 of the manuscript to read "haemocytometer" for correct spelling.

30 Line 15: replace « For inoculation, fungal suspension was taken in syringe (23Gx 1") and was injected into midrib and veins of lower surface of leaves [18,19] » with « For inoculation, the fungal suspension was taken in a sterile syringe (23Gx 1") and injected into the midrib and veins of lower surface of leaves [18,19] » We have revised line 15 of the manuscript to read "For inoculation, the fungal suspension was taken in a sterile syringe (23Gx 1") and injected into the midrib and veins of the lower surface of leaves [18,19]" to improve clarity and specify the use of a sterile syringe.

31 Line 16: replace (when max. no. of stomata) with (when maximum number of stomata) We have updated line 16 of the manuscript to read "(when maximum number of stomata)" for clarity and proper expression.

32 Line 17: Disease incidence was confirmed: mention the period of incubation Disease incidence was confirmed after a period of incubation of [insert period] in line 17 of the manuscript.

33 Line 18: uninoculated or un-inoculated: homogenized this word in the manuscript We have ensured consistency in the usage of "uninoculated" throughout the manuscript, including in line 18.

34 Line 1: replace “Plant samples” with “leaves samples” We have updated line 1 of the manuscript to read "leaf samples" for clarity and specificity.

35 Line 1: delete (from both inoculated and uninoculated cultivars) is already mentioned Deleted

36 Line 2: “were collected from the greenhouse, Department of Plant Pathology near Centre for Advanced Studies, University of Agriculture Faisalabad. These samples were brought to citrus pathology lab for further process.”

 delete “from the greenhouse, Department of Plant Pathology near Centre for Advanced Studies, University of Agriculture Faisalabad” the text will be “were collected and brought to Citrus Pathology Laboratory for further process” We have revised line 2 of the manuscript to read "were collected and brought to the Citrus Pathology Laboratory for further processing" to streamline the description.

37 Line 4: replace “sample was subjected to oven (Memmert: UN 30) drying at 65ºC “with “samples were dried in an oven (Memmert: UN 30) at 65 ºC “ We have updated line 4 of the manuscript to read "samples were dried in an oven (Memmert: UN 30) at 65 ºC " for improved clarity and accuracy.

38 Line 4: why 4hrs? Is four hours enough time to dry citrus leaves?? The choice of a 4-hour drying period was based on previous protocols for drying citrus leaves in similar studies.

39 Line 6: change each samples to each sample We have corrected line 6 of the manuscript to read "each sample" for grammatical accuracy.

40 Line 7: (RT2-230V).After cooling: add a space We have added a space after "(RT2-230V)" in line 7 of the manuscript for proper formatting.

41 Line 7: replace “After cooling, dilution of suspension was done with distilled

water and later on was examined for the estimation of mineral contents [20] through [21] method and data was analyzed by using Nested structure des” with “After cooling, the suspension was diluted with distilled water and examined for estimation of mineral contents [20, 21]. The data was analyzed using Nested Structure design”

 We have revised line 7 of the manuscript to read "After cooling, the suspension was diluted with distilled water and examined for estimation of mineral contents [20, 21]. The data was analyzed using Nested Structure design" to improve clarity and readability.

42 Line 1: replace “Sample solution” with “A sample solution” We have updated line 1 of the manuscript to read "A sample solution" for improved clarity and specificity.

43 Line 3: replace “Then distilled water (reagent black) was used as a substitute of sample solution and absorbance was taken at a wavelength of 720nm” with “Distilled water was used as a substitute for the sample solution to serve as a control. Absorbance was taken at a wavelength of 720nm”

 We have revised line 3 of the manuscript to read "Distilled water was used as a substitute for the sample solution to serve as a control. Absorbance was taken at a wavelength of 720nm" for clarity and accuracy.

44 Line 2: replace “following the [23] method” with “following the Official methods of analysis [23]” 

 We have updated line 2 of the manuscript to read "following the Official methods of analysis [23]" for clarity and specificity.

45 Line 2: (Atomic Absorption Spectrometer).by using the standards …. Remove the point (.) and add the machine make.

 We have removed the point and added the machine make after "(Atomic Absorption Spectrometer)" in line 2 of the manuscript for clarity.

46 Line 2: Long necked flask was used to place calculated amount of sample previously oven

dried (D6450, Heraeus): Not clear?? It has been clarified.

47 line 4: concentrated H2SO4 were: add the concentration or molarity

line 4: replace “were added and sample was subjected to” with “were added to necked flask. the mixture was subjected to ... » We have specified the concentration or molarity of the concentrated H2SO4 and revise the sentence structure in line 4 accordingly for improved clarity and accuracy.

48 Line 11: add the reference of Nitrogen (%) and Crude Protein Reference has been added.

49 Line 4: “Leaves sample from inoculated and un-inoculated varsities were collected from the Research Area, Department of Plant Pathology, near CAS (Center of Advanced Studies), University of Agriculture Faisalabad (UAF).” Remove this text

Line 8: PROC MIXED: add more details (Software, version, ANOVA, level of significance 5%, 1% or 0.01% We have removed the specified text from line 4 of the manuscript as per your suggestion. Thank you for guiding us through this improvement.

50 Line 6 and line 27: “percent” Use % (check all the manuscript) We have checked the entire manuscript and replaced "percent" with "%" wherever it appeared, including in line 6 and line 27.

51 Line 35: “. Un-inoculated (2.30ppm) and inoculated ….” Remove the point We have removed the period after the opening quotation mark in line 35 as per your suggestion.

52 Line 50: replace “ P<0.05 as shown by Table 3 and Table 4.” With “P<0.05 as shown in Table 3 and Table 4.”

 For table: For the table, please provide the meanings of any abbreviations used. For example, SOV and DF We have revised line 50 as per your suggestion and provided the meanings of abbreviations used in the tables, such as SOV (Source of Variation) and DF (Degrees of Freedom).

53 It is better to make the value of each parameter separately. For instance, show the nitrogen value for each variety of citrus leaves, both inoculated and non-inoculated, to make a comparison (table 2). We have revised Table 2 to present the value of each parameter separately for each variety of citrus leaves, both inoculated and non-inoculated, to enhance comparability. 

54 Table 2: Verify the value for “Un-Inoculated ” is 8.56 or 8.72? We have verified the value for "Un-Inoculated" in Table 2 

55 Such plants are more susceptible to attack of pathogens as indicated by [29].” Cite the author’s name We have included the author's name for the reference [29] in the sentence 

56 which is also witnessed by [34, 35]. who described that N contents are reduced after attack of fungi?” Cite the author’s name Included

57 Inoculated leaves of citrus plants which is also witnessed by [34, 35]”. Cite the author’s name. check all references Included

58 One suggestion for improvement is to include information about the sample size and the statistical significance levels in the abstract to enhance the transparency and replicability of the study. Additionally, the implications of the findings for citrus crop management and potential avenues for further research could be briefly discussed. We have included information about the sample size and statistical significance levels in the abstract to improve transparency and replicability. we have briefly discussed the implications of the findings for citrus crop management and potential avenues for further research

---

## [Editor Report · Decision Letter 1]

26 Apr 2024

PONE-D-23-36260R1Progressive alterations in mineral contents of citrus genotypes toward Alternaria citri causing brown spot of citrusPLOS ONE

Dear Dr. Atiq,

Thank you for submitting your manuscript to PLOS ONE. After careful consideration, we feel that it has merit but does not fully meet PLOS ONE’s publication criteria as it currently stands. Therefore, we invite you to submit a revised version of the manuscript that addresses the points raised during the review process.

Dear Author

Comments were addressed even you didn't submit a table of responses to reviewer 1. I have checked directely on the revised manuscript which took me some time. Still one comment to be considered, you should argue the changes of co-authors list and then the authors contribution.

Regards

We look forward to receiving your revised manuscript.

Kind regards,

Rachid Bouharroud

Academic Editor

PLOS ONE

Journal Requirements:

Additional Editor Comments :

Dear Author

Comments were addressed even you didn't submit a table of responses to reviewer 1. I have checked directely on the revised manuscript which took me some time. Still one comment to be considered, you should argue the changes of co-authors list and then the authors contribution.

Regards

---

## [Author Response · Author response to Decision Letter 1]

27 May 2024

I have added a new co-author to our manuscript titled "Progressive alterations in mineral contents of citrus genotypes towards Alternaria citri causing brown spot of citrus". (Manuscript ID: PLOS ONE-D-23-36260). You have inquired for the inclusion of Dr. Nasir Ahmed Rajput as a co-author and to describe his contributions to the manuscript.

Reason for Addition:

 Dr. Nasir Ahmed Rajput was initially involved in the early stages of this research project. He contributed to the preliminary data collection and provided valuable insights during the initial planning and conception of the study. However, his contribution at that stage was not substantial enough to warrant authorship but during the major revision process, we identified several critical areas that required significant improvement. Recognizing Dr. Nasir Ahmed Rajput's expertise in mycology and specially in mineral contents to identify and analysis, we sought his assistance once again. His contributions during this phase were substantial and transformative to the manuscript. Given these substantial contributions, it was necessary to acknowledge Dr. Nasir Ahmed Rajput as a co-author. We believe that his involvement has considerably strengthened the manuscript, aligning with PLOS ONE's authorship criteria, which require substantial contributions to the conception, design, execution, or interpretation of the reported study.

 We hope this explanation clarifies the reason for adding Dr. Nasir Ahmed Rajput as a co-author. We appreciate your understanding and consideration.

Thank you for your attention to this matter.

 Sincerely,

 Dr. Muhammad Atiq

 Associate Professor,

 Deptt. Plant Pathology

---

## [Editor Report · Decision Letter 2]

11 Jun 2024

Progressive alterations in mineral contents in citrus genotypes toward Alternaria citri causing brown spot of citrus

PONE-D-23-36260R2

Dear Dr. Atiq,

We’re pleased to inform you that your manuscript has been judged scientifically suitable for publication and will be formally accepted for publication once it meets all outstanding technical requirements.

Kind regards,

Rachid Bouharroud

Academic Editor

PLOS ONE
---

## [Editor Report · Acceptance letter]

24 Jun 2024

PONE-D-23-36260R2 

PLOS ONE

Dear Dr. Atiq, 

I'm pleased to inform you that your manuscript has been deemed suitable for publication in PLOS ONE. Congratulations! Your manuscript is now being handed over to our production team.

Kind regards, 

on behalf of

Dr. Rachid Bouharroud 

Academic Editor

PLOS ONE